# Field-Based Biomechanical Assessment of the Snatch in Olympic Weightlifting Using Wearable In-Shoe Sensors and Videos—A Preliminary Report

**DOI:** 10.3390/s23031171

**Published:** 2023-01-19

**Authors:** Cheng Loong Ang, Pui Wah Kong

**Affiliations:** Physical Education and Sports Science Academic Group, National Institute of Education, Nanyang Technological University, Singapore 637616, Singapore

**Keywords:** barbell, velocity, trajectory, ground reaction force, power, kinetic, kinematic

## Abstract

Traditionally, the biomechanical analysis of Olympic weightlifting movements required laboratory equipment such as force platforms and transducers, but such methods are difficult to implement in practice. This study developed a field-based method using wearable technology and videos for the biomechanical assessment of weightlifters. To demonstrate the practicality of our method, we collected kinetic and kinematic data on six Singapore National Olympic Weightlifters. The participants performed snatches at 80% to 90% of their competition one-repetition maximum, and the three best attempts were used for the analysis. They wore a pair of in-shoe force sensors loadsol^®^ (novel, Munich, Germany) to measure the vertical ground reaction forces under each foot. Concurrently, a video camera recorded the barbell movement from the side. The kinematics (e.g., trajectories and velocities) of the barbell were extracted using a free video analysis software (Kinovea). The power–time history was calculated from the force and velocity data. The results showed differences in power, force, and barbell velocity with *moderate* to *almost perfect* reliability. Technical inconsistency in the barbell trajectories were also identified. In conclusion, this study presented a simple and practical approach to evaluating weightlifters using in-shoe wearable sensors and videos. Such information can be useful for monitoring progress, identifying errors, and guiding training plans for weightlifters.

## 1. Introduction

The sport of Olympic weightlifting consists of two competition lifts, the snatch and the clean and jerk. The winner of a competition is determined by the highest amount of weight lifted by each weightlifter. Olympic weightlifting is a power sport wherein the two competition lifts, the snatch and the clean and jerk, are placed in the middle of the force–velocity curve [1]. The sport demands a high level of both physical ability and technical proficiency from the athletes. Therefore, higher amounts of power output will directly translate to better weightlifting performance [2,3,4,5], and technical proficiency can be quantified through the profiling of the barbell trajectory and the extent of horizontal displacement [6,7,8,9,10,11]. A routine assessment of the kinetics and kinematics variables is therefore of great importance for coaches to obtain performance data pertaining to a weightlifter’s current competitive condition, assisting with decision making on future training modalities.

The focus of this study was on the snatch, which consists of six phases: first pull, transition, second pull, turnover, and recovery [12,13,14]. Snatch performance has been related to various biomechanical parameters, such as the velocity, force, power, and trajectory of the barbell. A high barbell velocity [15,16,17] and the ability to produce high amounts of force [5,18,19] are important contributing factors to successful snatch performance. As power is the product of velocity and force, an increase in force production and velocity will directly contribute to power production. Power has been shown to be useful in predicting weightlifting performance, as the sport requires a high amount of force to be generated in a short amount of time [20,21,22]. Therefore, obtaining the power profile of a weightlifter provides crucial information on their current performance [23,24].

Traditionally, the assessment of weightlifting performance variables has been carried out in a biomechanics laboratory with the use of force platforms to measure the ground reaction forces under the feet and a position transducer attached to the barbell to obtain the displacement and velocity profile [24,25]. While laboratory-grade equipment provides detailed and accurate measurements of weightlifting movements, this assessment approach is not accessible to most coaches and athletes. Video-based methods can be a cost-effective alternative to a position transducer. For, example, the use of a mobile application on a smartphone device or the use of sports analytics software on a captured video have been shown to be reliable and well-validated methods for the measurement of velocity and barbell trajectory [26,27,28,29]. In the field of biomechanics, devices such as portable force plates and wearables force sensors have been developed to facilitate force measurement outside of a laboratory setting [30,31,32,33]. The advancement of wearable technology and video analysis presents new opportunities to develop a portable solution for measuring the kinetics and kinematics of Olympic weightlifting.

Thus, the objective of this study was to develop a field-based method for comprehensively assessing the biomechanical profiles of weightlifting athletes. Ideally, this method should be portable, requiring a short setup time, and easily deployable in a field setting. The cost of the field-based method will be much lower compared to a laboratory setup. The biomechanical information obtained could be useful for monitoring progress, identifying errors, and guiding training plans for weightlifters.

## 2. Materials and Methods

### 2.1. Participants

This study was approved by Nanyang Technological University Institutional Review Board (reference number: IRB-2022-491). After providing written informed consent, the participants filled in a background survey form and an inclusion/exclusion criteria form to assess their eligibility to participate in the study. The exclusion criteria for the study were: weightlifters who were not a part of the Singapore National Olympic Weightlifting team; aged below 21 or above 40; or unable to perform a snatch due to pain, injury, or pregnancy at the time of the study. All six participants (3 males and 3 females) were national-level athletes in the Singapore national team at the time of the study (Table 1). They were in good health to perform a snatch.

### 2.2. Equipment

Wireless loadsol^®^ insoles (novel, Munich, Germany) were used to measure the vertical ground reaction force under each foot of the weightlifters during their snatch attempts. The loadsol^®^ insoles have been well-validated as a reliable tool for measuring ground reaction force in sporting activities such as running [33,34], jumping, and landing [35,36]. They have already been applied in research to measure plantar pressure distribution in Olympic weightlifters [37]. Compared with portable force platforms, the loadsol^®^ in-shoe sensors offer advantages such as being lightweight, portable, and easy to set up. The loadsol^®^ has a compatible application (loadsol-s) for both iOS and Android devices that can be paired with the loadsol^®^. In this study, the loadsol-s (Version 1.7.53) Android application (Figure 1a) was used for data acquisition and the extraction of the force–time data in ASCII format.

A Nikon DLSR D3500 camera with an 18–55 mm zoom lens attached was used to capture the snatch attempts, and the videos were shot in 1080 p resolution with 60 frames per second at a shutter speed of 1/120 s. The camera was placed facing the left side of the weightlifter 3 m away from the platform (Figure 1b). An LED emitter was attached to the end of a barbell to be used as a marker to facilitate auto-tracking of the trajectory using Kinovea [29]. A meter stick was placed in alignment with the edge of the barbell where the LED emitter was attached, providing a reference measure for the conversion of pixels to meters in the video analysis.

### 2.3. Experimental Procedures

The collection of data was carried out at the training ground of the Singapore Weightlifting Federation, where the participants usually trained. The body mass and height of the participants were measured. The shoe sizes of the weightlifters were obtained prior to the data collection. A pair of size-matched loadsol^®^ insoles were assigned to the participants and inserted into their shoes. For loadsol^®^ calibration, the zeroing of the insole was achieved through the unloading of the left foot followed by the right foot. The weight for the snatch attempts was 80% to 90% of the weightlifters’ competition one-repetition maximum. This weight range has been observed to produce optimal load and power spectra, resulting in representative maximum power outputs for weightlifters [38,39,40]. The protocols were rehearsed during the warmup to the actual test weight for the snatch attempts. A maximum of ten attempts were performed, and the best three among the successful snatch attempts were used for the data analysis. The attempts chosen for the analysis were successful lifts performed with good control and consistency, as indicated by the participants and the researcher. Three minutes of rest time were allowed in between each attempt to reduce fatigue, helping the weightlifters maintain good technical and power efficiency [41,42].

In order to calculate the power parameters, we had to obtain force and velocity data concurrently. However, the loadsol^®^ (for force measurements) and the video camera (for velocity measurements) were two independent systems with no synchronization capability between them. To match the timings of both systems, the weightlifters were instructed to jump and land right before their snatch attempt while both systems were recording the data. The purpose of this jump was to create a key event to facilitate the manual synchronization of data extracted from the loadsol^®^ and the videos. When the weightlifter was in the airborne phase of the jump, the loadsol^®^ forces were near zero. The instant at which a sharp rise occurred in the measured forces due to the weightlifter landing the jump was identified from the force–time data in ASCII format extracted using the loadsol-s Android application. Any excess data before this point were trimmed off. The video frame in which the participant’s foot first touched the ground was identified by navigating through the video frame by frame with the use of Kinovea software. Similarly, the excess video before this point was trimmed off. The data of both systems were then synchronized based on the time point at which the foot made contact with the floor as the weightlifter landed the jump. The manual synchronization process was repeated to find the starting point of the measurement (i.e., the moment of separation of the barbell from the floor) as well as the terminal point of the measurement (i.e., the point at which the peak velocity occurred). All excess data after the terminal point were trimmed off, retaining only the data of interest. This manual synchronization process aligned the force–time and velocity–time histories, making it possible to calculate the power–time history using data obtained from the two separate measuring systems.

### 2.4. Data Processing

Ground reaction force data were exported in ASCII format using the loadsol-s Android application. Barbell displacement and velocity data were obtained via video analysis in Kinovea (version 0.9.5, Figure 1b). Kinovea is a free open-source two-dimensional (2D) motion analysis software whose reliability for the analysis of biomechanics measurements such as motion- and time-related variables in multiple sports, including Olympic Weightlifting, has been well-validated [28,29,43,44,45,46,47]. The tracing of the snatch trajectory was carried out using the auto-tracking function of Kinovea with the tracker placed at the edge of the barbell where the LED emitter was attached. The X and Y coordinates and the horizontal displacement of the snatch attempts were extracted. Ground reaction force and the barbell kinematic data were analyzed according to the three acceleration phases of the snatch [48]. Starting from the moment of separation of the barbell from the floor during the first pull, when the inertia of the barbell has to be overcome, the first pull is a more strength-orientated phase due to the lower barbell velocity [16,49]. The transition phase lies in between the first and second pull, when a temporary loss in ground reaction force occurs due to the double knee bend [50,51]. The last phase is the second pull, consisting of the explosive and forceful extension of the knees and hips and ankle plantarflexion, when the peak velocity of the barbell is reached, making the second pull a more power-orientated phase [14,15,16,49] (Figure 2). The barbell starts decelerating after reaching peak vertical velocity; therefore, the data obtained after reaching the peak vertical velocity were not taken into account [2,52]. Three key positions were identified within the snatch trajectory: the first pull, the second pull, and the descent of the barbell after the barbell reached maximal vertical displacement. The key positions were used to assess the amount of horizontal displacement of the barbell in reference to the vertical reference line. A negative displacement value represents the barbell moving away from the weightlifter, while a positive value represents the barbell moving toward the weightlifter [13,17,53].

The exported data were further processed using a customized MATLAB code. The vertical displacement data of the barbell were filtered through a 12 Hz low-pass Butterworth filter and then differentiated to calculate the vertical velocities. From the synchronized velocity–time and force–time data, we calculated the power–time histories of the barbell (Figure 2). Key data extracted for analysis included peak and mean velocity, peak and mean force, and peak and mean power measures. Several other parameters, such as the time to peak velocity, the barbell trajectory, and the maximum horizontal displacement of the barbell, were obtained directly from the video analysis using Kinovea.

### 2.5. Reliability Analysis

To check the consistency among the 3 trials per participant, reliability analyses of peak velocity, peak force, and peak power were conducted using statistical procedures. Intra-class correlation coefficients (ICC(2,1)) were computed in SPSS (version 29, IBM Corp., Armonk, SA, USA). The ICCs were interpreted as slight (<0.20), fair (0.21–0.40), moderate (0.41–0.60), substantial (0.61–0.80), or almost perfect reliability (>0.80) [54]. The associated standard error of measurement (SEM) was calculated from the ICC results and the standard deviation (SD) of the data.

## 3. Results

### 3.1. Velocity

The barbell velocity measurements of the three snatch attempts performed by each of the six participants were observed to be consistent, and there was little deviation between each individual’s attempts (Table 2). Interestingly, one female weightlifter (F1) had both the lowest mean velocity and the highest peak velocity and took the longest time to reach peak velocity. Despite the males lifting heavier weights than the females, both sexes displayed similar peak velocity values. Some weightlifters displayed two peaks in velocity during their snatch attempt, with a noticeable dip in velocity observed during the transition phase of the snatch (Figure 3). Most participants had a much steadier increase in velocity, as illustrated by the representative profile shown in Figure 2b.

### 3.2. Force

A clear sex difference in force production could be observed (Table 2). While participant M3 had the lowest mean and peak force production, he displayed the most consistent peak force values among the male participants. In general, all female participants individually had consistent force production for three of their snatch attempts, with participant F3 having the highest mean and peak forces.

### 3.3. Power

The male participants had similar mean powers but varied results for peak power (Table 2). Participant M3 had a much lower peak power compared to participant M2, who competes in a similar weight category and lifted a similar amount of weight for the test (Table 1). Participant M1, who competes in a lighter 67 kg weight category, also demonstrated a higher peak power value compared to participant M3. In terms of the consistency between the three attempts, participant M3 showed the lowest variation in peak power production.

Regarding the power output observed among the females, participant F3 had both the highest peak and mean power. While participant F1 had a higher peak power than participant F2, the opposite was found for mean power. When considering the body weight of the participants, participant F1, who competes in the lighter weight category, had the highest peak power relative to her body weight, and participant F3 was a close second.

### 3.4. Barbell Trajectory

The barbell trajectories revealed different styles of snatch techniques (Figure 4). The snatch trajectories of participant M1 crossed the vertical reference line twice, while participant M2 had a backward displacement without crossing the reference line. The remaining participants M3, F1, F2, and F3 crossed the reference line thrice. Participants M1 and F2 displayed the most consistent snatch trajectories among all participants.

All the weightlifters were consistent in their first pull, with some weightlifters preferring to pull the barbell towards themselves and others away. There was more variability between the weightlifters during the second pull and the descent from max height to the catch position. Participants M2, M3, and F1 had the least consistency in their horizontal displacement during the descent. M2 displayed backward horizontal displacement, with a positive value in all three key positions in all of his snatch attempts (Table 2). 

### 3.5. Reliability Analysis

The ICC analysis showed that the reliability was *almost perfect* for the peak force and peak power measurements and *moderate* for peak velocity (Table 3). The associated measurement errors are presented in Table 3.

## 4. Discussion

This study set out to develop a practical field-based method for the biomechanical assessment of Olympic weightlifters using wearable technology and videos. Key kinetic (e.g., force and power) and kinematic (e.g., trajectory and velocity) parameters were successfully obtained from in-shoe force sensors and video tracking using free software. The results could provide coaches and sports scientists with insights into the current competitive level of weightlifters and guide their training plans.

### 4.1. Biomechanical Profiles of the Snatch

Among the six national athletes who participated in this study, considerable variation in the biomechanical profiles of the snatch were observed. For instance, female weightlifter F1 had the lowest mean velocity but the highest peak velocity. This could be attributed to the much higher time to peak velocity, causing a decrease in the mean velocity. Among the males, participants M1 and M3 had similar peak force values, despite M1 competing in the lighter weight category, whereas M2 had the highest peak force among the male weightlifters. M3 had the highest peak velocity among the male weightlifters but, due to his lower force production, had a much lower peak power compared to M2, who competes in a similar weight category, and M1, who competes in a lighter weight category.

Sex differences in the data were more apparent in terms of force and power production, with male weightlifters observed to produce approximately twice the amount of force compared to female weightlifters. This was not surprising, because male weightlifters are stronger than female weightlifters within similar competitive levels [55]. The sex difference in strength is observed to occur after the age of 10 years old, when boys reach puberty and their performance increases rapidly, widening the strength difference between the sexes. Body mass is also a contributing factor to strength differences, as athletes with a higher body mass are generally much stronger and able to lift more weight than their lighter counterparts [56]. Therefore, maximum strength is a major contributing factor for Olympic weightlifting due to the large amount of force that is required to overcome the weight loaded on the barbell, and it is hence a strong indicator of Olympic weightlifting performance [5,18,19,57,58].

The velocity data were much closer between the sexes, which could be attributed to the difference in the weight loaded onto the barbell during the snatch attempts. Barbell velocity decreases as the weight on the barbell increases, which is known as the load–velocity relationship. The loss in velocity has been attributed mostly to the first pull of the snatch [48,59]. The similar velocities between the sexes could also be due to the athletes reaching the threshold velocity, defined as the minimum amount of vertical velocity required to pull the bar high enough in order to drop down and catch the bar, which can differ for each individual weightlifter. An increase in force production helps improve the load–velocity relationship while fulfilling the threshold velocity required for the weightlifter to get under the barbell, even with heavier weights being lifted [48]. It is also important to note that an excessive amount of velocity is suboptimal due to the waste of force that could have been used to lift heavier weights [50]. Although the velocity values were similar among the participants, some weightlifters displayed a linear increase in barbell velocity, including a more skillful weightlifter (Figure 2b). In contrast, a less skillful weightlifter displayed two velocity peaks (Figure 3), with a loss in velocity during the transition phase of the snatch, which could be attributed to technical flaws [11,16,48]. During the transition, an unweighting phase occurs due to the double knee bend, defined as the re-engagement of the knee extensor, which can help establish quadriceps muscular pre-tension for the second pull [60,61]. The double knee bend can cause a temporary decrease in the ground reaction force (Figure 2c), but a more skillful weightlifter can minimize the loss in velocity during the transition phase of the snatch [50,51]. The power output of the female weightlifters was much closer to that of the male weightlifters in comparison to force production. Since Olympic weightlifting is a strength–speed sport, and power is the product of force and velocity, this could be useful in predicting weightlifting performance [4,5]. Therefore, the biomechanical profiles of velocity, force, and power that were measured using the field-based method presented in this study could provide crucial information for assessing a weightlifter’s ability to produce force and maintain an optimal vertical barbell velocity to meet the threshold velocity at a heavier load.

According to Vorobyev (1978) [62], the barbell trajectory of a snatch is typically an S-shaped curve. Three basic snatch technique types have been categorized based on the characteristics of the trajectories in relation to the vertical reference line drawn from the starting position of the barbell. The Type 1 trajectory crosses the vertical reference line twice, the Type 2 trajectory has a backward displacement without crossing the reference line, and the Type 3 trajectory crosses the reference line thrice. In the present study, all three types of trajectory were observed in the Singapore National Olympic Weightlifters (Figure 4). M1 displayed a Type 1 trajectory, and M2 displayed a Type 2 trajectory, which is the most frequently observed trajectory in elite weightlifters [6,8,63]. The remaining participants (M3 and all three female weightlifters) displayed a Type 3 trajectory. Our observations were in alignment with the findings of Hiskia (1997) [64], who observed that a high proportion of female weightlifters displayed a Type 3 trajectory. The trajectory of F3 had a curvature during the descent to the catch position (Figure 4) due to a clockwise rotational motion in the transverse plane, which is common and can be caused by asymmetrical development in the weightlifter [65]. Differences in the physical attributes of each weightlifter can influence the trajectory type displayed [63,66]. It has been suggested that weightlifters with longer trunks display Type 2 and 3 trajectories, while weightlifters with longer lower extremities display a Type 1 trajectory more frequently [39]. There is no consensus on the optimal barbell trajectory for a snatch, but consistency and the minimization of horizontal displacement leading to less energy wastage from the stabilization of the barbell are desired [6,49,63,67]. The field-based approach presented in this study provides coaches with a practical method to assist with the regular assessment of the snatch performance of weightlifters, providing crucial performance information.

Consistency while performing a snatch may be crucial to success, as more skilled weightlifters display more consistent and stable movement patterns while minimizing the extent of horizontal displacement [49,63,67]. The maximum horizontal displacement observed at the three key positions of the snatch performed by a weightlifter can provide some insight into the consistency of their technique. Inconsistency in the trajectory was more apparent in the second pull and the descent. Overall, participants M1 and F2 displayed high consistency throughout the three key positions compared to the other weightlifters. It is, however, important to note that a successful snatch has multiple determinants, and that an optimal barbell trajectory has not been determined [6,63].

### 4.2. Comparison with Existing Methods

The existing laboratory methods for determining the kinematic and kinetic profiles of weightlifting movements often involve the use of a force platform, a position transducer, or both [24]. In our study, we used wearable in-shoe force sensors and video analysis to determine similar kinematic and kinetic variables. The new method proposed in this study drastically improves the ease of conducting biomechanical testing outside of a laboratory setting.

The use of video analysis in sport is not new, and this approach has already been implemented for the collection of kinematic variables related to Olympic weightlifting movements. The kinematics data and barbell trajectory profiles measured in the present study were comparable to those reported in other studies using video analysis [7,10,11,12,13,17,52,63,68,69]. On the other hand, the use of in-shoe force sensors for the measurement of ground reaction forces generated from weightlifting movements is relatively new. The kinetic variable values obtained herein were also comparable to those of other studies using either calculated values or direct measurements from force platforms [2,11,17,39,52]. In addition to being portable and lightweight, in-shoe force sensors inherently offer the advantage of enabling the assessment of potential bilateral kinetic asymmetry in weightlifters by comparing the ground reaction forces recorded in the left and right insoles. While a similar bilateral comparison can also be achieved with the use of two synchronized force platforms, this laboratory test approach involves a considerably higher cost and more complex setup [70,71,72]. Previous studies have suggested that asymmetry in power production and lean mass may decrease performance and increase the chance of injury in athletes [70,71,72,73]. The use of in-shoe force sensors as a method to assess the kinetic asymmetry of Olympic weightlifters could be further explored in the future.

### 4.3. Limitations

This study had several limitations. First, the use of 2D kinematics measurements has its disadvantages, since the data obtained only consider movements in the sagittal plane and not the possible rotational movement in the transverse plane. The use of 3D motion capture [12,69,74,75] would allow the attainment of a more comprehensive kinematics profile [44] but would require a more complicated setup with multiple cameras, which is not practical for field-based assessments. Overall, 2D analysis remains an effective, low-cost, and convenient option given the ready availability of video-capturing devices such as smartphones or consumer cameras paired with free software such as Kinovea [44]. Second, consumer cameras are limited by relatively low frame rates. The present study recorded the snatch movements at 60 Hz, which was deemed sufficient for the chosen movements. High-speed cameras, coupled with fast shutter speeds and sufficient lighting, can capture high-quality video at a higher frame rate [68,69,76]. Lastly, three of the participants (M1, M3, and F1) competed in the Birmingham 2022 Commonwealth Games just before the data collection commenced, and they may have been tired from traveling and competing. The differences in the competitive schedules of the participants could have influenced the data; hence, the results may not accurately reflect the participants’ abilities [77,78,79]. Moving forward, it would be useful to regularly assess athletes’ performance throughout their training and competition phases via longitudinal monitoring. In future research, integrating multiple wearable devices with smartphones via sports analytics applications could be the way forward. This would offer a mobile solution to comprehensively assess the motion and forces involved in Olympic weightlifting, including bilateral asymmetry between the left and right sides. Although this field-based setup is not as accurate as laboratory equipment, it can provide sufficient data for the day-to-day assessment of athletes, with the added advantage of the ease of use and availability of smartphones [80].

## 5. Conclusions

This study developed a field-based biomechanical assessment method for profiling the kinetics and kinematics of the snatch in Olympic weightlifting. The method is practical and easy to set up, owing to the convenience of the loadsol^®^ in-shoe force sensors in conjunction with video analysis via the free software Kinovea. The results identified differences in power and force production between male and female weightlifters. The biomechanical variables showed *moderate* to *almost perfect* reliability across repeated trials. The new method presented in this study helped assess the weightlifters’ ability to maintain the required threshold velocity for a successful snatch and potential technical flaws due to a loss in velocity at the transition phase. It could also help coaches and scientists to identify technical inconsistencies at certain phases of the snatch. Given the portability of the developed method, the profiling of weightlifters could be performed regularly for longitudinal monitoring. Such information could help trainers assess the effectiveness of training, identify errors, and guide future training and competition plans. In conclusion, the field-based method developed in this study is viable for assessing the biomechanical profiles of the snatch movement in Olympic weightlifting.

## Figures and Tables

**Figure 1 sensors-23-01171-f001:**
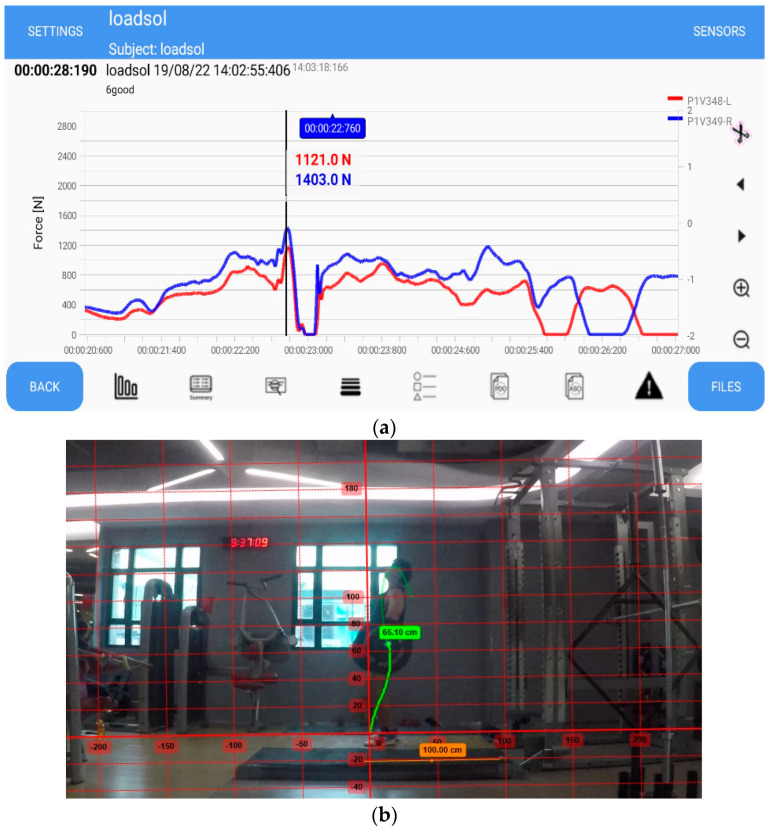
(**a**) Screenshot of a force−time profile recorded using the loadsol−s Android application; (**b**) screenshot of video tracking of barbell trajectory using Kinovea freeware.

**Figure 2 sensors-23-01171-f002:**
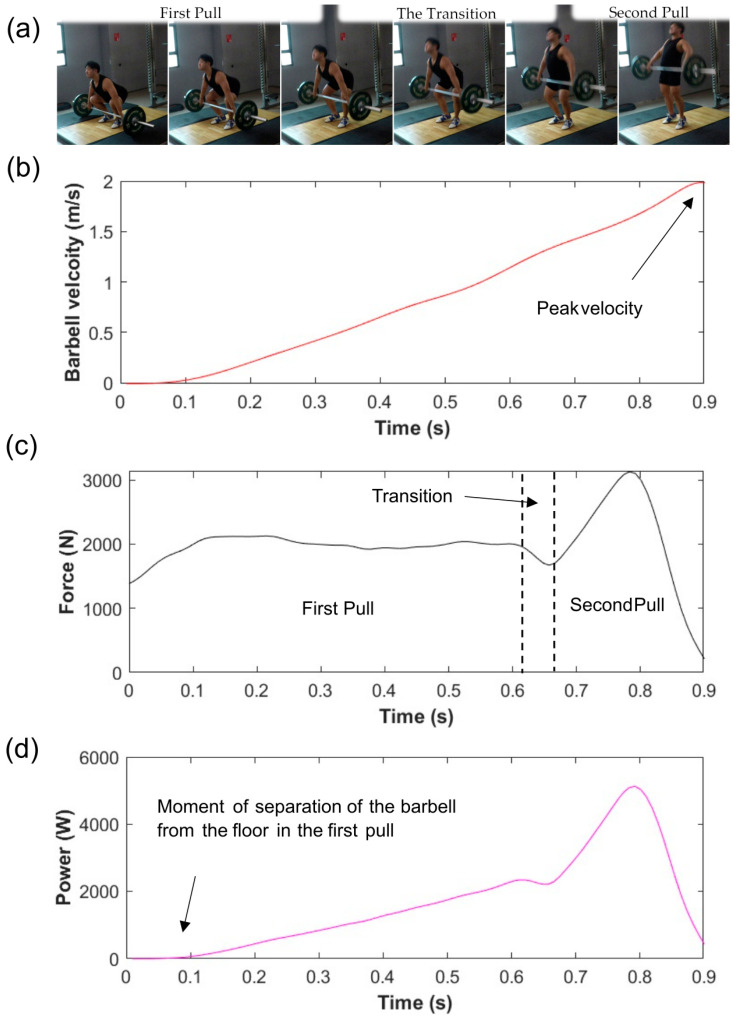
(**a**) Representative pictures of the three snatch phases and associated (**b**) velocity–, (**c**) force–, and (**d**) power–time graphs from the moment of separation of the barbell in the first pull to the moment at which peak velocity occurred in the second pull of the snatch.

**Figure 3 sensors-23-01171-f003:**
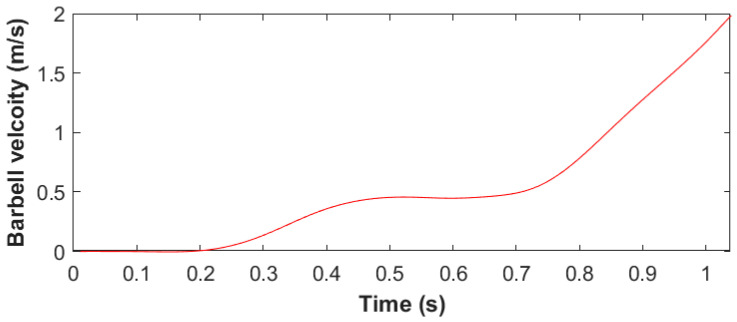
Example of a velocity–time graph profile with two peaks for participant F1 while performing a snatch.

**Figure 4 sensors-23-01171-f004:**
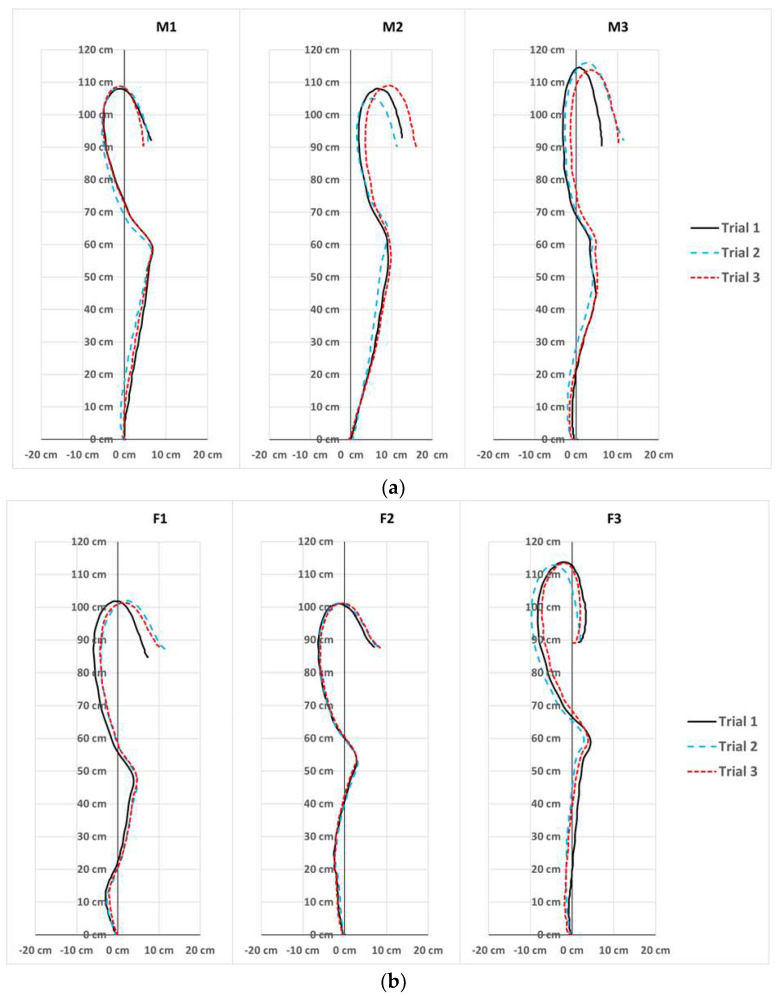
Barbell trajectory graphs plotted using the X and Y coordinates of the three snatch attempts performed by each of the six National Olympic Weightlifters: (**a**) male weightlifters and (**b**) female weightlifters.

**Table 1 sensors-23-01171-t001:** Participant characteristics of male and female weightlifters in the Singapore national team.

	Participants					
	M1	M2	M3	F1	F2	F3
Age (years)	24	24	26	27	23	27
Competition weight category (kg)	67	81	81	49	55	55
Body mass (kg)	69.4	80.5	82.0	49.9	56.0	55.5
Weight of snatch attempts (kg)	86	105	106	48	48	48
Weightlifting experience (years)	2	1	4	3	1.5	4

M = male, F = female.

**Table 2 sensors-23-01171-t002:** Snatch performance variables of the male and female weightlifters.

	Participants					
	M1	M2	M3	F1	F2	F3
Mean velocity (m/s)	0.88 ± 0.07	0.77 ± 0.05	0.87 ± 0.03	0.61 ± 0.02	0.85 ± 0.05	0.90 ± 0.09
Peak velocity (m/s)	2.01 ± 0.02	1.99 ± 0.11	2.11 ± 0.02	2.15 ± 0.02	2.06 ± 0.04	2.10 ± 0.02
Time to peak velocity (s)	0.87 ± 0.04	0.93 ± 0.06	0.91 ± 0.03	1.07 ± 0.01	0.80 ± 0.03	0.86 ± 0.07
Mean force (N)	1827.49 ± 54.73	1982.03 ± 33.17	1731.64 ± 39.19	997.35 ± 14.43	964.86 ± 30.39	1340.14 ± 59.18
Peak force (N)	2553.01 ± 237.91	3025.94 ± 270.98	2535.98 ± 59.95	1656.86 ± 46.78	1455.12 ± 64.72	1846.41 ± 69.70
Mean power (W)	1587.59 ± 26.18	1598.96 ± 81.19	1533.94 ± 61.57	647.33 ± 94.61	818.68 ± 62.02	1198.06 ± 150.08
Peak power (W)	4119.24 ± 339.01	5116.71 ± 509.05	3899.42 ± 119.90	2830.67 ± 477.57	2408.27 ± 173.53	3144.45 ± 132.42
Mean power relative to bodyweight (W/kg)	22.88	19.86	18.71	12.99	14.62	21.61
Peak power relative to bodyweight (W/kg)	59.36	63.56	47.55	56.78	43.00	56.71
Max horizontal displacement, first pull (cm)	2.43 ± 0.64	5.32 ± 0.47	−1.53 ± 0.58	−2.53 ± 0.45	−2.42 ± 0.19	−1.30 ± 0.48
Max horizontal displacement, second pull (cm)	−5.20 ± 0.21	2.29 ± 1.11	−2.42 ± 1.01	−4.68 ± 0.99	−6.09 ± 0.28	−8.31 ± 1.27
Max horizontal displacement Descent from max height (cm)	5.53 ± 0.88	13.38 ± 2.55	9.30 ± 2.77	9.75 ± 2.24	7.79 ± 0.74	2.60 ± 0.67

M = male, F = female.

**Table 3 sensors-23-01171-t003:** Reliability statistics of the measurements.

Performance Variable	ICC	95% Confidence Interval	SEM
Peak velocity	0.567	[0.028, 0.915]	0.05 m/s
Peak force	0.939	[0.785, 0.990]	155.3 N
Peak power	0.894	[0.648, 0.983]	334.7 W

ICC = intraclass correlation coefficient, SEM = standard error of measurement.

## Data Availability

Data are available at the NIE Data Repository (https://doi.org/10.25340/R4/CVVUOX).

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
