# Peer review of "Field-Based Biomechanical Assessment of the Snatch in Olympic Weightlifting Using Wearable In-Shoe Sensors and Videos—A Preliminary Report"

_sensors, 2023, doi:10.3390/s23031171_

Round 1

Reviewer 1 Report

The authors have submitted an interesting manuscript that uses pressure insoles and video analysis for the purpose of studying weightlifting biomechanics. In general, I like the study and think the authors did a nice job. That said, there are a few issues that need to be addressed:

1. There is too much focus on power throughout the manuscript. Please note that while weightlifters display high levels of power during the exercises that does not mean more power is better. For example, the velocity only has to be a certain threshold value to reach the height needed for the lifter to drop underneath - any additional velocity is wasted. The lifter should focus on lifting heavier weights instead. Please see Baumann et al. 1988 for discussion (your reference 11). There are also several editorials in the Journal of Sport Science and The Journal of Strength and Conditioning Research that actively discourage researchers from using power as a variable. So, again I would suggest the authors edit the manuscript to minimize the focus on power.

2. The methods section is lacking some detail about how the phases were defined. It is also not clear exactly how the onset of the lift was defined. Based on some the data in the Figures I am concerned that some of the GRF/loadsol data are not trimmed correctly to coincide with the actual lift. As an example, in Figure 4a there is not discernible beginning. The authors should consider including vertical or horizontal lines (e.g., like in reference 11 or by Enoka) to show when the lift starts and ends.

2.1 I am not sure the data presented in Figure 2 are accurate i.e., the linear increase in velocity is not typical for skilled weightlifters (and this looks more like bar position). I am also concerned about the time course for the Force data in Figure 2b - would the bar velocity really increase until there is no more ground force applied? It seems like the velocity should be decreasing at that point already. Again, perhaps this is actually bar height? Please check the data. Also, including the phases and clearly defining them in the methods would help clarify this figure, too.

3. Please use the same software program for all Figures. More consistency among the Figures would greatly improve the manuscript.

4. For section 4.2 there is no actual data presented in the manuscript. The authors only show one Figure to illustrate that asymmetries were present, but the reader has no idea of the magnitude or how consistent these asymmetries were across lifters. It would ne nice to have more data on this. Otherwise, I would suggest deleting this section since there is a lack of data to support the discussion.

5. Given the purpose "to develop a field-based method" the authors should consider calculating ICCs or technical errors etc. This would help practitioners know how accurate the values in Table 2 are and would help them understand what would be a meaningful change.

6. The discussion focuses a lot on kinematic results e.g., barbell velocity and trajectory types. But that was only half of the data collected. The authors should consider discussing the force data in greater detail, too. Again, see earlier comments about GRF phase definition, which may help with this, too.

I feel the authors have an interesting manuscript and I hope they take the time to make revisions and improve the manuscript. I look forward to seeing the next submission. Good luck.

Round 2

Reviewer 1 Report

The authors have done a good job adressing my comments and concerns. Overall I feel that the manuscript is much improved. There are only two minor issues that I feel the authors should consider:

1. The Methods section still lacks a bit of detail about how the GRF and kinematic data were synchronized. In addition, the reference (48) provided and the "three acceleration phase" model does not provide information about GRF phase definitions - I see that the authors can use the video data to visually identify the beginning, but the loadsol does not have video, correct? At least the screenshot does not show video. Again, just some clarification would be nice.

2. I appreciate that the authors improved the consistency in their Figures, but Figure 4 does not look like it was made with MATLAB. Again, this is a minor issue and I would leave it up to the authors to decide if they want to use the same program for all Figures.
